# Estimated Phytate Intake Is Associated with Bone Mineral Density in Mediterranean Postmenopausal Women

**DOI:** 10.3390/nu15071791

**Published:** 2023-04-06

**Authors:** Pilar Sanchis, Rafael María Prieto, Jadwiga Konieczna, Félix Grases, Itziar Abete, Jordi Salas-Salvadó, Vicente Martín, Miguel Ruiz-Canela, Nancy Babio, Jesús Francisco García-Gavilán, Albert Goday, Antonia Costa-Bauza, José Alfredo Martínez, Dora Romaguera

**Affiliations:** 1Laboratory of Renal Lithiasis Research, University Institute of Health Science Research (IUNICS), Health Research Institute of the Balearic Islands (IdISBa), University of the Balearic Islands, 07122 Palma de Mallorca, Spain; rafelm.prieto@uib.es (R.M.P.); fgrases@uib.es (F.G.);; 2Consorcio CIBER, M.P. Fisiopatología de la Obesidad y Nutrición (CIBERObn), Instituto de Salud Carlos III (ISCIII), 28029 Madrid, Spain; jadzia.konieczna@gmail.com (J.K.); iabetego@unav.es (I.A.); jordi.salas@urv.cat (J.S.-S.); mcanela@unav.es (M.R.-C.); nancy.babio@urv.cat (N.B.); jalfmtz@unav.es (J.A.M.); 3Research Group on Nutritional Epidemiology & Cardiovascular Physiopathology (NUTRECOR), Health Research Institute of the Balearic Islands (IdISBa), 07120 Palma de Mallorca, Spain; 4Department of Nutrition, Food Sciences, and Physiology, Center for Nutrition Research, Navarra Institute for Health Research (IdiSNA), University of Navarra, 31008 Pamplona, Spain; 5Unitat de Nutrició Humana, Departament de Bioquímica i Biotecnologia, Universitat Rovira i Virgili, 43007 Tarragona, Spain; 6Institut d’Investigació Pere Virgili (IISPV), 43204 Reus, Spain; 7CIBER Epidemiología y Salud Pública (CIBERESP), Instituto de Salud Carlos III (ISCIII), 28029 Madrid, Spain; vicente.martin@unileon.es; 8Institute of Biomedicine (IBIOMED), University of León, 24071 León, Spain; 9Department of Preventive Medicine and Public Health, IdiSNA, University of Navarra, 31008 Pamplona, Spain; 10Cardiovascular Risk and Nutrition Research Group, Hospital del Mar Medical Research Institute (IMIM), Department of Medicine, University of Barcelona, 08193 Barcelona, Spain; agoday@parcdesalutmar.cat; 11Precision Nutrition and Cardiometabolic Health Program, IMDEA Food, CEI UAM + CSIC, 28049 Madrid, Spain

**Keywords:** phytate, bone mineral density, DXA, postmenopausal women

## Abstract

The main objective of this work was to explore the association of dietary phytate intake with bone mineral density (BMD) in a Mediterranean population of postmenopausal women. For this purpose, a cross-sectional analysis of 561 women aged 55–75 years with overweight/obesity and metabolic syndrome from a Mediterranean area and with data on dual-energy X-ray absorptiometry (DXA) scans in femur and lumbar spine was performed. Estimated phytate intake was calculated using a validated food frequency questionnaire. Our results indicated that phytate intake was associated with BMD [β(95%CI) per each 25 mg/100 kcal] in femoral neck [0.023(0.060–0.040) g/cm^2^], femoral Ward’s triangle [0.033(0.013–0.054) g/cm^2^], total femur [0.018(0.001–0.035) g/cm^2^], and all the analyzed lumbar spine sites [L1–L4: 0.033(0.007–0.059) g/cm^2^] after adjusting for potential confounders. The sensitivity analysis showed that phytate intake was directly associated with lumbar spine BMD in women younger than 66 years, with a body mass index higher than 32.6 kg/cm^2^ and without type 2 diabetes (all *p*-for interactions < 0.05). The overall results indicated that phytate, a substance present in food as cereals, legumes and nuts, was positively associated with BMD in Mediterranean postmenopausal women. Phytate may have a protective effect on bone resorption by adsorbing on the surfaces of HAP. Nevertheless, large, long-term, and randomized prospective clinical studies must be performed to assess the possible benefits of phytate consumption on BMD in postmenopausal women.

## 1. Introduction

Osteoporosis is a bone disease that develops when bone mineral density (BMD) and bone mass decrease, or when the quality and/or structure of bone changes. This can lead to a decrease in bone strength that can increase the risk of fractures, among the major public health concerns in the elderly population [1,2]. Several risk factors seem to play a role in osteoporosis development. The amount of bone mass accumulated from childhood to early adulthood is among the most important predictors of osteoporosis risk later in life [3]. Nutrition and physical activity are the most important modifiable factors with a well-known effect on bone maintenance and mass loss [4].

Postmenopausal women are at increased risk of osteoporosis because the drop of estrogen during this period leads to more bone resorption than formation, which can result in osteoporosis [5]. It is widely accepted that a balanced diet helps maintain bone health [6,7] and is important for the prevention of osteoporosis in postmenopausal women. Several studies have shown that certain nutrients, especially vitamin D and calcium, are associated with BMD [5,6,7,8]. Additional research has evaluated the relationship between bone density and other nutrients, such as essential fatty acids and vitamins [9]. A high dietary glycemic index has also been reported as a risk factor of osteoporotic fractures [10]. In addition, some studies have shown that the lower incidence of osteoporosis in some Mediterranean countries may be related to their healthy diet [11,12,13]. The Mediterranean diet is characterized by high consumption of grains, legumes, vegetables, fruits, nuts, and olive oil. The useful results of a number of its components (along with vegetables, fish, and fruits) on BMD had been formerly studied [11,12,13,14,15].

Myo-inositol hexakisphosphate calcium magnesium salt or phytin is a salt of phytic acid, an inositol ring with six phosphate groups, that acts as a reservoir for phosphorus in seeds and plant germination [16]. Phytate is a natural compound that is consumed in significant amounts (in a range of 1–2 g/day) by people on a diet rich in legumes, whole grains, and nuts. Some authors have suggested that phytate exhibits effects similar to those of bisphosphonates on bone resorption [17,18,19,20,21]. In a study with female ovariectomized rats [17], BMD values were significantly higher in both femoral bones and L4 vertebra for phytate-treated rats in comparison to rats in the non-phytate group. Phytate has been correlated with bone mass in postmenopausal subjects in a small cohort study [18,19,20]. Additionally, it has been demonstrated that phytate can act as an osteoclastogenesis inhibitor in cell and tissue studies [21]. Another study has indicated that the phytate effect on decalcification process seems to resemble that of bisphosphonates, similar to alendronate and greater than etidronate [22]. In this sense, phytate may have positive effects on bone health, with a mechanism of action on bone resorption similar to that exhibited by bisphosphonates. It could be explained, almost in part, because they have a high affinity to bind onto the calcium of hydroxyapatite crystals by chemisorption, hindering both crystallization and redissolution. Additionally, a recent in vitro study indicates that phytate can inhibit or disturb the decalcification process by adsorbing on the surfaces of hydroxyapatite (HAP) crystal and the consequent inhibition of HAP dissolution [22] and osteoclast activity [21]. Nevertheless, larger studies are needed to confirm these findings.

Our hypothesis is that an adequate daily phytate intake (0.5–1 g/day) protects against bone mass loss and an association between dietary phytate and bone parameters will be detected in our population. So, the main objective of this work is to explore the association of the dietary phytate intake with bone parameters in a Mediterranean population of women.

## 2. Materials and Methods

### 2.1. Study Design and Population

A cross-sectional analysis in a subset of women from the PREvención con DIeta MEDiterránea Plus (PREDIMED-Plus) trial was performed. The trial’s design and methods used have been described previously [23,24]. The trial design and methods are available at http://predimedplus.com (accessed on 3 April 2018). In summary, PREDIMED-Plus is a 6-year, randomized, parallel-group, multicenter, controlled study on primary prevention of cardiovascular disease that is ongoing in Spain. The trial objective is to evaluate the effect of an intensive weight-loss intervention with an energy-reduced Mediterranean diet, a behavioral support on the prevention of cardiovascular events and physical activity promotion, in comparison to usual care. The trial includes 6874 men and women, aged 55–75 with overweight/obesity (body mass index (BMI) ≥ 27 kg/m^2^ and <40 kg/m^2^) and who met at least three characteristics of the metabolic syndrome [25]. For the present study, we have included a subset of 561 women with data on bone parameters in femur and lumbar spine at baseline, coming from 4 of the 23 PREDIMED-Plus recruiting centers (7 PREDIMED-Plus centers had access to dual-energy X-ray absorptiometry scanners and 4 of them performed measurements of BMD in addition to total body composition).

### 2.2. Estimated Phytate Intake and Other Nutritional Variables

Estimation of phytate intake was previously described [26]. It is based on the determination of consumption of the major food sources of phytate (legumes, whole-cereals, and nuts). These were collected with a validated semi-quantitative food frequency questionnaire (FFQ) composed of 143 items [27,28], considering the serving size of each item [28], and the phytate proportion of each item based on published sources [29,30,31,32,33]. Table 1 shows estimated phytate content for standard serving sizes, based on the reported phytate content for selected items in the FFQ. FFQ data were used to determine consumption of specific food groups (vegetables and fruits (g/day) and Spanish food composition tables [34,35] were used to derive data on total energy (kcal/day) and micronutrients (calcium (mg/day), vitamin D (µg/day)) intake.

Using average values from the International Tables of Glycemic Index [36] and glucose as the reference food, the glycemic index was estimated for each FFQ item. The glycemic index was assessed as the glycemic load of the diet divided by the grams of total carbohydrates consumed per day and expressed as percentage.

The FFQ collected information about the food intake during the previous year. The FFQ is repeated yearly in PREDIMED-Plus participants but, given that this is a cross-sectional study using baseline data, only data of the first FFQ—administered at recruitment—is presented.

### 2.3. Bone Parameters’ Assessment

At baseline, bone parameters were measured by trained operators using third-generation dual-energy X-ray absorptiometry (DXA) scanners from General Electric (DXA Lunar Prodigy Primo and Lunar iDXA; GE Healthcare, Madison, WI, USA) connected with enCoreTM software. BMD was determined at the non-dominant femur (neck, Ward’s triangle, trochanter, diaphysis and total) and lumbar spine in anterior–posterior position (L1–L2, L1–L3, L1–L4, L2–L3, L2–L4, and L3–L4), and expressed in g/cm2 and in standard deviations from the young adult normal mean values (T-score), based on the Spanish reference population provided by manufacturer. In case of diaphysis, the T-score values were not available. In addition, we categorized women into two groups based on modified T-score cut-offs established by the World Health Organization (WHO) for each area [37]: low-BMD when T-score was equal or lower than −1 and normal-BMD in another case. The standard WHO categories of BMD state (normal, osteopenia, and osteoporosis) were modified due to low total number of osteoporotic cases. DXA scans, including subject positioning and daily phantom calibration, were performed following manufacturer guidelines.

### 2.4. Assessment of Other Variables

At baseline, trained dieticians collected information on socio-demographics, lifestyle habits and health status using general questionnaire. To categorize educational level (further education/technician, secondary education/primary education or less), and smoking behavior (never, former, current), three groups were created. Baseline prevalence of type 2 diabetes (T2D) and self-reported osteoporotic fractures was used as a dichotomous variable (yes/no). The BMI was calculated as weight (kg)/height (m) squared. Initial weight, in light clothing, and height were collected by qualified personnel using calibrated scales. The average value was utilized for analysis. Total leisure-time physical activity (MET•min/week) was determined using the Minnesota-REGICOR short physical activity questionnaire, previously validated in the Spanish population [38].

### 2.5. Statistical Analysis

The subset of women was separated into three groups based on their phytate tertiles: low (T1: <15 mg/100 kcal), moderate (T2: 15.0–28.4 mg/100 kcal) and high (T3: >28.4 mg/100 kcal). Unless otherwise specified, data are provided as numbers and percentages, means and standard deviations, or means and standard errors. The chi-square test was used for categorical variables, and one-way ANOVA and LSD were used as post hoc tests for quantitative data in intergroup comparisons of phytate groups.

Linear and logistic regression models were fitted to assess the associations between phytate intake with BMD expressed as absolute values (g/cm^2^) and T-score values (linear regression models) or low BMD status (logistic regression). For these analyses, we used tertiles of phytate intake in both linear and logistic models, considering the first tertile (low phytate intake) as the reference category. Multivariate models were adjusted for age (years), BMI (kg/m^2^), physical activity (MET•min/week), educational level (higher education/technician or secondary education/primary education or less), smoking status (never/former/current), T2D prevalence, self-reported osteoporotic fractures, intake of total energy (kcal/day), calcium (mg/day), vitamin D (µg/day), glycemic index, and consumption of vegetables and fruits (g/day).

Sensitivity analyses were conducted using the T-scores for lumbar spine L1–L4 measures as the outcome: these included effect modification analyses and stratification of participants by the median of age (≤66 y/>66 y), BMI (≤32.6 kg/m^2^/>32.6 kg/m^2^) and T2D prevalence (yes/no) groups. Models were adjusted by the same variables used in the main analyses. All graphs and tests yielded models that met the independence of observations, homogeneity of variance, and normality of residuals criteria.

For these analyses, we used the official PREDIMED-Plus database generated on the 22 December 2020. A two-tailed *p*-value less than 0.05 was considered statistically significant. Statistical analyses were performed using SPSS 23.0 (SPSS Inc., Chicago, IL, USA).

## 3. Results

A total of 561 women were included in the present study with a mean age of 66.5 ± 4.1 years and BMI of 33.0 ± 3.6 kg/cm^2^. The main anthropometric, dietary and lifestyle data of participants are shown in Table 2. Women in the highest tertile of phytate intake had higher physical activity and a lower glycemic index than those in the lowest tertile. Regarding food intake, women in the top tertile consumed more vegetables, legumes and nuts and less meat, olive oil and pastries and sweets when compared to the lowest tertile.

Table 3 shows the BMD values (expressed in g/cm^2^ and T-scores) according to tertiles of phytate intake. Women in the top tertile had higher BMD values in almost all sites, both expressed as absolute values or T-scores, compared to those in the lowest tertile. Estimated phytate intake (mg/100 kcal) was also statistically higher for women with normal BMD (T-score > −1) compared to those with low BMD (T-score < −1) for Ward’s femoral triangle, lumbar spine L1–L3, L1–L4, L2–L3 and L2–L4 sites (Appendix A).

We further explored the association between phytate intake and BMD (g/cm^2^) values in univariate and multivariate linear regression analysis (Table 4). After adjusting for potential cofounders, phytate intake (by tertiles and per each 25 mg/100 kcal) was directly and significantly associated with BMD in femoral neck, femoral Ward’s triangle, total femur and all the analyzed lumbar spine sites but not in case of femoral trochanter (*p* = 0.064) and femoral diaphysis (*p* = 0.072).

Table 5 shows the beta-coefficients and 95% CI of the associations between phytate intake (by tertiles and per each 25 mg/100 kcal) and BMD T-scores values. As can be observed, after adjusting for potential confounders, phytate intake was positively and significantly associated with T-scores in femoral neck, femoral Ward’s triangle, and all the analyzed lumbar spine sites but not in case of total femur and femoral trochanter.

Binary logistic regression was also performed to study the association between phytate intake and low BMD defined as a T-score ≤ 1 (Appendix A). After adjusting for potential cofounders, phytate intake was inversely associated with low BMD status in femoral Ward’s triangle, lumbar spine L1–L3 and lumbar spine L2–L3 sites but not in the rest of the analyzed sites (Figure 1).

Finally, we carried out stratified analyses by age, BMI and T2D for the association between phytate intake and the T-score in lumbar spine L1–L4, which is the most common measure used in the diagnosis of osteopenia or osteoporosis in lumbar spine (Table 6). Phytate intake was associated with T-score values in women younger than 66 years, with a BMI higher than 32.6 kg/cm^2^ and with no T2D (*p* for all interactions < 0.05).

## 4. Discussion

The present study reported a positive association between phytate intake and BMD in a sample of postmenopausal women with overweight/obesity and metabolic syndrome from Spain.

Our results are in accordance with previous observational studies that indicate that phytate intake has a positive association with bone health. Body weight and low phytate intake were risk factors for BMD of the lumbar vertebrae and neck of the femur in a cross-sectional investigation of postmenopausal women [20]. In another study in postmenopausal women, the 10-year fracture probability in the low-phytate group was significantly higher than in the high-phytate group, both in major osteoporotic and hip fracture [18]. Phytate, in an in vitro study, inhibited hydroxyapatite (HAP) dissolution in a concentration-dependent manner by adsorbing phytate in HAP sur-faces, and this effect was comparable to alendronate but greater than that of etidronate [22]. Moreover, it has been shown that phytate inhibits osteoclastogenesis in human primary osteoclast cell line and in RAW 264.7 monocyte/macrophage mouse cell line [21]. Last but not least, it has been noted that phytate consumption produces phytate hydrolysates (InsP5, InsP4, InsP3, InsP2) by intestinal phosphatase activity [30] that may also have a significant impact on bone resorption by adsorbing in crystal surfaces of HAP [39]. Altogether, phytate may therefore offer, by adhering to HAP surfaces and reducing osteoclast activity, a unique type of nutraceutical or therapeutic agent of the decalcification process.

The bisphosphonate is currently among the most popular therapies for osteoporosis to lower the risk of fracture and reduce bone resorption. These are effective drugs for bone disorders characterized by increased bone resorption, such as Paget’s disease, osteoporosis, hypercalcemia of cancer, multiple myeloma, and bony metastases [40,41]. The bisphosphonates are adsorbed very effectively to HAP, the crystalline form of calcium and phosphate in bone. Bisphosphonates are analogues of pyrophosphate which have potent inhibitory effects on bone resorption. Pyrophosphate is an important endogenous body’s regulator of calcification. Within humans, pyrophosphate is released as a product of many synthetic reactions, and it has been detected in many tissues, including blood and urine [42,43]. However, it was demonstrated that pyrophosphate could inhibit pathological calcification only when it was injected rather than ingested. For this reason, the use of pyrophosphate as a drug was constrained by its pharmacokinetics because it was hydrolyzed and inactivated when was administered orally [42,43]. Consequently, bisphosphonates were found to be chemically stable analogues of pyrophosphate that could inhibit pathological calcification and bone resorption when they were administered orally [44,45]. Similar to their natural analogue pyrophosphate, bisphosphonates have a very high affinity for bone mineral because they bind to hydroxyapatite crystals. Therefore, the retention of bisphosphonates on bone surface depends on the availability of hydroxyapatite binding sites. To date, many studies using in vitro systems, animal models, and clinical trials have shown that a variety of bisphosphonates can inhibit bone resorption [44,45].

Similar to bisphosphonates and pyrophosphate, phytate can act as inhibitor of both calcification and decalcification processes. In this sense, phytate has also demonstrated to inhibit the formation of pathological calcifications (such as renal calculi [39,46,47], dental calculi [48], and cardiovascular calcification [49,50,51]). Moreover, some studies in animals and cells indicate that phytate may also provide protection against cancer [52], Parkinson’s disease [53], cognitive degeneration [54] and diabetes-related diseases [55,56]. Phytate is natural polyphosphate readily accessible to people consuming a balanced diet (0.5–1 g/day) through nuts, legumes and cereals. Interestingly, the Mediterranean diet results in consumption of approximately 1 g of phytate/day [26]. All mammalian fluids, tissues, and organs contain phytate [57,58], and its contents rely on the exogenous supply either orally [57,58] or topically [59,60]. In this way, after 22 days without phytate in the diet, the urine content of phytate is undetectable [57,58]. It is understood that phytate’s capacity to form complexes with iron, zinc, copper, magnesium, and calcium may reduce these minerals’ bioavailability from food [57]. Foods of plant origin have a high phytate content expressed as magnesium, potassium and calcium salts. Together, all these salts are called phytates, but they have a different solubility and ability to complex other divalent cations [30,61]. In this sense, the different phytate salts can modify the bioavailability of the different ions such as calcium, iron and zinc to a different extent [30,62,63]. For this reason, phytate has been labeled as anti-nutrients that chelate metal ions and thus reduce their bioavailability. It is important to remark that this antinutritional effect only occur in unbalanced diets when the mineral intake is very low and dietary phytate is very high [30,64]. In this sense, some authors have indicated that this negative effect on mineral bioavailability may appear in growing children [65], adolescents and pregnant mothers who consume diets rich in phytates (cereal-based foods) and low in minerals [32,66]. Almost all of these cases are reported in low-income countries [67] as a consequence of the adherence to diets that do not agree with the mineral recommended intakes. So, under non-varied and non-balanced dietary conditions, phytate may affect the bioavailability of iron, zinc and calcium [30,32,62,63,64,65,66,67]. Nevertheless, no negative effects on mineral status have been observed in mineral balanced diets (as Mediterranean diet) with adequate amounts of phytate (0.5–1 g/day) [26,57,58,59,60]. Furthermore, a recent paper has indicated that a phytate consumption higher than 307 mg/day was associated with a normal lumbar BMD (t-score > −1) in postmenopausal women [22]. Consequently, a healthy and balanced diet with the right amounts of these trace minerals may prevent the loss of bone mass and prevent trace metals deficiencies [57,58,59,60].

Our study has some limitations, the first is the cross-sectional nature, which precludes conclusions regarding the temporal nature of our findings and no causality can be established. Even though we found that phytate intake is associated with BMD, we cannot confirm which one is the cause and which is the effect; and we cannot confirm that a higher phytate consumption is associated with a lower risk of fracture or osteoporosis. For these reasons, prospective studies are needed to establish the time sequence in the relationship between them and clinically relevant findings. The second limitation is that our participants are Mediterranean postmenopausal women with over-weight/obesity and metabolic syndrome, and our results cannot be generalized to other populations. Another limitation is that phytate intake is an “estimated measure”, although it has been calculated as previously described [26] based on the FFQ consumption data. On the other hand, the major strengths are the use of DXA scan for BMD determination, the control for many potential cofounders and the inclusion of sensitivity analyses.

## 5. Conclusions

In conclusion, our results show that phytate intake, a natural product present in legumes, nuts and whole cereals, is positively associated with BMD in Mediterranean postmenopausal women.

We hypothesize that eating a diet rich in phytate may prevent or alleviate disorders that result in bone mass loss, such as osteoporosis. The mechanism of action of phytate on bone resorption could be explained by its capacity to adsorb on the surfaces of hydroxyapatite (HAP) crystal and the consequent inhibition of HAP dissolution [22] and osteoclast activity [21]. However, to evaluate the impact of phytate consumption on BMD, large, lengthy, and randomized prospective clinical trials must be carried out.

## Figures and Tables

**Figure 1 nutrients-15-01791-f001:**
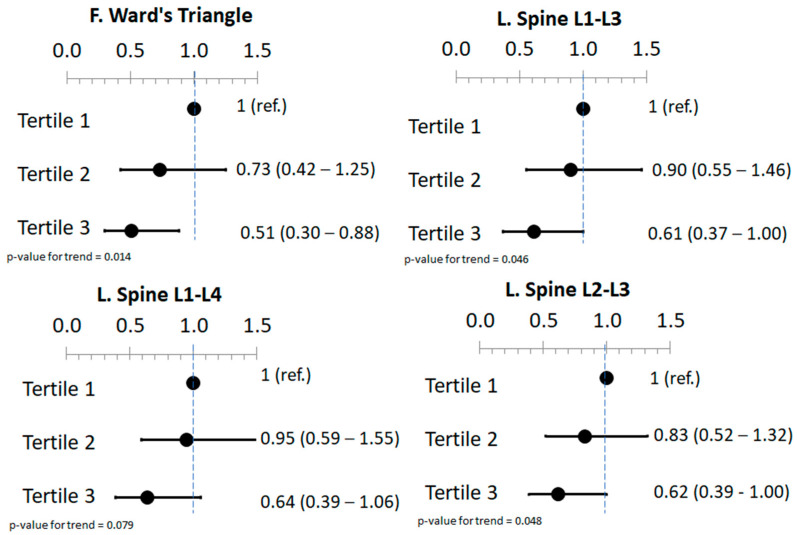
Forest plot of low bone mineral density (T-score ≤ −1) and tertiles of estimated phytate intake. Values are expressed as the adjusted odds ratio (95% CI). Models were adjusted for age (years), BMI (kg/m^2^), physical activity (MET•min/week), educational level (higher education/technician or secondary education/primary education or less), smoking status (never/former/current), type 2 diabetes prevalence, osteoporotic fractures prevalence, energy (kcal/day), calcium (mg/day), vitamin D (µg/day), glycemic index, vegetables and fruits (g/day). This figure is a summary of the Appendix A.

**Table 1 nutrients-15-01791-t001:** Estimated phytate content for standard serving sizes, based on the reported phytate content for selected items in the food frequency questionnaire (FFQ) [26,27,28,29,30,31,32,33,34,35].

Food	Estimate mg Phytate/100 g Edible	Serving Size (g)	Phytate per Serving (mg)
Green beans	180	200 *	360
Almonds, peanuts, hazelnuts, pistachio or pine seed	1000	30	300
Walnuts	1600	30	480
Lentils	400	150 *	600
Beans (pinto, kidney or lima)	700	150 *	1050
Chickpeas	400	150 *	600
Beans and broadbeans	600	150 *	900
Whole bread	350	75	263
Whole cereals (muesli, oatmeal, all-bran)	350	30	90
Whole rice	350	60	210
Wholemeal cookies	300	50	150
Whole-wheat pasta	300	60	180

* Weight after cooking.

**Table 2 nutrients-15-01791-t002:** Anthropometric, lifestyle and dietary characteristics of study participants according to tertiles of estimated phytate intake (mg/100 kcal·day).

	Tertile 1		Tertile 2		Tertile 3		
	<15.0 mg/100 kcal		[15.0–28.4] mg/100 kcal		>28.4 mg/100 kcal		
Variable	(n = 187)		(n = 187)		(n = 187)		*p*-Value
Age, years	66.3 ± 4.1		67.1 ± 3.9		66.2 ± 4.1		0.062
BMI, kg/m^2^	33.4 ± 3.5		32.5 ± 3.3 ^a^		33.1 ± 3.8		0.046
Physical activity, MET•min/week	1709 ± 1602		2389 ± 2153 ^a^		2331 ± 1738 ^a^		<0.001
Menopausal age, years	48.7 ± 5.6		49.4 ± 5.3		48.6 ± 6.3		0.387
Educational level, n (%)							
Higher education	22	(11.8%)	20	(10.8%)	28	(15.2%)	0.341
Technician or secondary education	43	(23.1%)	40	(21.5%)	50	(27.2%)	
Primary education or less	121	(65.1%)	126	(67.7%)	106	(57.6%)	
Smoking status, n (%)							
Never	119	(64.0%)	132	(70.6%)	122	(65.2%)	0.126
Former	44	(23.7%)	43	(23.0%)	53	(28.3%)	
Current	23	(12.4%)	12	(6.4%)	12	(6.4%)	
Type 2 diabetes, n (%)	47	(25.1%)	46	(24.6%)	48	(25.7%)	0.972
Osteporotic fractures, n (%)	4	(2.1%)	5	(2.7%)	2	(1.1%)	0.523
Nutrients							
Total energy intake, kcal/day	2376 ± 520		2338 ± 630		2250 ± 576		0.099
Phytate, mg/100 kcal	9.4 ± 3.2		21.1 ± 4.0 ^a^		40.5 ± 9.8 ^a,b^		<0.001
Phytate, mg/day	225 ± 98		490 ± 155 ^a^		912 ± 331 ^a,b^		<0.001
Vitamin D, µg/day	5.8 ± 3.3		6.3 ± 3.3		6.6 ± 3.5		0.057
Calcium, mg/day	1067 ± 353		1084 ± 377		1037 ± 366		0.454
Phosphorous, mg/day	1731 ± 419		1812 ± 479		1877 ± 458 ^a^		0.008
Zinc, mg/day	12.7 ± 3.1		13.2 ± 3.4		14.1 ± 3.8 ^a,b^		0.001
Glycemic index	55.5 ± 4.4		53.7 ± 5.0 ^a^		52.4 ± 6.0 ^a^		<0.001
Food							
Vegetables (g/day)	300 ± 109		344 ± 126 ^a^		354 ± 131 ^a^		<0.001
Fruits (g/day)	362 ± 217		371 ± 238		353 ± 180		0.701
Legumes (g/day)	18 ± 8		21 ± 11 ^a^		21 ± 13 ^a^		0.007
Cereals (g/day)	153 ± 81		144 ± 78		152 ± 78		0.520
Whole cereals (g/day)	11 ± 22		43 ± 38 ^a^		123 ± 84 ^a,b^		<0.001
Dairy (g/day)	395 ± 205		371 ± 216		363 ± 218		0.335
Meat (g/day)	159 ± 52		154 ± 59		140 ± 54 ^a,b^		0.004
Olive oil (g/day)	45 ± 15		42 ± 15		40 ± 15 ^a^		0.007
Fish (g/day)	100 ± 43		104 ± 44		104 ± 45		0.530
Nuts (g/day)	5 ± 8		15 ± 15 ^a^		25 ± 21 ^a,b^		<0.001
Pastries and sweets (g/day)	34 ± 37		27 ± 35 ^a^		21 ± 25 ^a^		0.001

Values are expressed as the mean ± SD. The *p*-value was calculated with ANOVA test using LSD in the post hoc analysis. ^a^: *p* < 0.05 vs. T1; ^b^: *p* < 0.05 vs. T2.

**Table 3 nutrients-15-01791-t003:** Bone mineral density values (g/cm^2^ and T-score) according to tertiles of estimated phytate intake (mg/100 kcal·day).

Variable	Tertile 1 <15.0 mg/100 kcal	Tertile 2 [15.0–28.4] mg/100 kcal	Tertile 3 >28.4 mg/100 kcal	*p*-Value
BMD g/cm^2^				
Femoral Neck, g/cm^2^	0.86 ± 0.11	0.86 ± 0.12	0.90 ± 0.12 ^a,b^	0.004
Femoral Ward’s Triangle, g/cm^2^	0.66 ± 0.11	0.67 ± 0.13	0.71 ± 0.16 ^a,b^	0.000
Femoral Trochanter, g/cm^2^	0.78 ± 0.10	0.77 ± 0.11	0.80 ± 0.12 ^b^	0.014
Femoral Diaphysis, g/cm^2^	1.16 ± 0.14	1.16 ± 0.15	1.19 ± 0.16 ^a,b^	0.038
Total Femur, g/cm^2^	0.95 ± 0.10	0.95 ± 0.12	0.98 ± 0.13 ^a,b^	0.010
Lumbar Spine L1–L2, g/cm^2^	1.02 ± 0.15	1.01 ± 0.15	1.07 ± 0.18 ^a,b^	0.001
Lumbar Spine L1–L3, g/cm^2^	1.06 ± 0.16	1.05 ± 0.15	1.11 ± 0.17 ^a,b^	0.002
Lumbar Spine L1–L4, g/cm^2^	1.07 ± 0.16	1.08 ± 0.16	1.12 ± 0.17 ^a,b^	0.012
Lumbar Spine L2–L3, g/cm^2^	1.09 ± 0.18	1.09 ± 0.16	1.13 ± 0.18 ^b^	0.018
Lumbar Spine L2–L4, g/cm^2^	1.10 ± 0.18	1.11 ± 0.17	1.15 ± 0.18 ^b^	0.042
Lumbar Spine L3–L4, g/cm^2^	1.12 ± 0.19	1.14 ± 0.18	1.17 ± 0.19	0.054
BMD T–scores				
Femoral Neck	−1.00 ± 0.87	−1.01 ± 0.99	−0.71 ± 1.03 ^a,b^	0.003
Femoral Ward’s Triangle	−1.82 ± 0.87	−1.79 ± 1.03	−1.49 ± 1.26 ^a,b^	0.005
Femoral Trochanter	−0.09 ± 0.92	−0.20 ± 1.00	0.05 ± 1.07	0.061
Total Femur	−0.28 ± 0.84	−0.38 ± 0.99	−0.13 ± 1.03 ^a,b^	0.045
Lumbar Spine L1–L2	−1.21 ± 1.23	−1.31 ± 1.22	−0.77 ± 1.47 ^a,b^	0.001
Lumbar Spine L1–L3	−0.96 ± 1.31	−1.01 ± 1.24	−0.53 ± 1.45 ^a,b^	0.002
Lumbar Spine L1–L4	−0.91 ± 1.34	−0.86 ± 1.33	−0.48 ± 1.44 ^a,b^	0.009
Lumbar Spine L2–L3	−0.91 ± 1.49	−0.93 ± 1.33	−0.54 ± 1.50 ^ab^	0.018
Lumbar Spine L2–L4	−0.83 ± 1.49	−0.75 ± 1.41	−0.44 ± 1.50 ^a^	0.033
Lumbar Spine L3–L4	−0.65 ± 1.57	−0.54 ± 1.50	−0.26 ± 1.56	0.054

BMD: Bone mineral density. Values are expressed as the mean ± SD. The *p*-value was calculated with ANOVA using LSD in the post hoc analysis. ^a^: *p* < 0.05 vs. T1; ^b^: *p* < 0.05 vs. T2.

**Table 4 nutrients-15-01791-t004:** Association of bone mineral density (g/cm^2^) with estimated phytate intake (by tertiles and per each 25 mg/100 kcal).

	Tertile 1 <15.0 mg/100 kcal	Tertile 2 [15.0–28.4] mg/100 kcal	Tertile 3 >28.4 mg/100 kcal	*p*-Value for Trend	Phytate (per 25 mg/100 kcal)	*p*-Value
Femoral Neck, n	186	187	183		556	
Crude Model	0 (reference)	−0.002 (−0.026–0.022)	0.034 (0.010–0.058)	0.005	0.026 (0.009–0.043)	0.003
Adjusted Model *	0 (reference)	0.003 (−0.021–0.026)	0.031 (0.007–0.056)	0.011	0.023 (0.006–0.040)	0.008
Femoral Ward’s Triangle, n	186	187	183		556	
Crude Model	0 (reference)	0.005 (−0.023–0.032)	0.050 (0.022–0.077)	<0.001	0.035 (0.016–0.055)	<0.001
Adjusted Model *	0 (reference)	0.009 (−0.019–0.037)	0.048 (0.020–0.077)	0.001	0.033 (0.013–0.054)	0.001
Femoral Trochanter, n	186	187	183		556	
Crude Model	0 (reference)	−0.011 (−0.033–0.011)	0.022 (0.000–0.044)	0.055	0.018(0.002–0.034)	0.027
Adjusted Model *	0 (reference)	−0.006 (−0.027–0.016)	0.020 (−0.002–0.043)	0.075	0.015 (−0.001–0.031)	0.064
Femoral Diaphysis, n	178	184	180		542	
Crude Model	0 (reference)	0.000 (−0.031–0.031)	0.035 (0.004–0.066)	0.026	0.023 (0.001–0.045)	0.041
Adjusted Model *	0 (reference)	0.009 (−0.021–0.039)	0.033 (0.002–0.063)	0.035	0.020 (−0.002–0.041)	0.072
Total Femur, n	178	184	180		542	
Crude Model	0 (reference)	−0.004 (−0.028–0.019)	0.030 (0.006–0.054)	0.015	0.021 (0.004–0.038)	0.015
Adjusted Model *	0 (reference)	0.003 (−0.020–0.026)	0.027 (0.004–0.051)	0.023	0.018 (0.001–0.035)	0.034
Lumbar Spine L1–L2, n	148	157	164		469	
Crude Model	0 (reference)	−0.012 (−0.047–0.023)	0.052 (0.017–0.088)	0.003	0.043 (0.018–0.067)	0.001
Adjusted Model *	0 (reference)	−0.004 (−0.039–0.032)	0.043 (0.007–0.080)	0.016	0.035 (0.010–0.060)	0.006
Lumbar Spine L1–L3, n	147	157	164		468	
Crude Model	0 (reference)	−0.007 (−0.043–0.029)	0.050 (0.014–0.086)	0.005	0.040 (0.015–0.065)	0.002
Adjusted Model *	0 (reference)	0.001 (−0.036–0.037)	0.042 (0.005–0.079)	0.003	0.033 (0.008–0.059)	0.011
Lumbar Spine L1–L4, n	147	158	167		472	
Crude Model	0 (reference)	0.005 (−0.032–0.043)	0.050 (0.013–0.087)	0.007	0.038 (0.013–0.063)	0,003
Adjusted Model *	0 (reference)	0.013 (−0.024–0.050)	0.045 (0.007–0.083)	0.019	0.033 (0.007–0.059)	0.013
Lumbar Spine L2–L3, n	167	171	176		514	
Crude Model	0 (reference)	−0.003 (−0.040–0.034)	0.044 (0.008–0.081)	0.017	0.033 (0.007–0.059)	0,012
Adjusted Model *	0 (reference)	0.006 (−0.031–0.043)	0.041 (0.006–0.083)	0.032	0.030 (0.003–0.056)	0.028
Lumbar Spine L2–L4, n	167	173	179		519	
Crude Model	0 (reference)	0.008 (−0.029–0.046)	0.045 (0.007–0.082)	0.018	0.033 (0.007–0.060)	0.012
Adjusted Model *	0 (reference)	0.017 (−0.021–0.055)	0.044 (0.006–0.083)	0.023	0.032 (0.005–0.058)	0.019
Lumbar Spine L3–L4, n	167	170	176		513	
Crude Model	0 (reference)	0.013 (−0.026–0.053)	0.047 (0.008–0.086)	0.019	0.036 (0.008–0.063)	0.012
Adjusted Model *	0 (reference)	0.022 (−0.018–0.062)	0.047 (0.006–0.087)	0.025	0.034 (0.005–0.062)	0.020

Linear regression models were used to evaluate the association between bone mineral density and estimated phytate intake (tertiles and per each 25 mg/100 kcal). Results are expressed as β coefficients (95% CIs). * Models adjusted for age (years), BMI (kg/m^2^), physical activity (MET•min/week), educational level (higher education/technician or secondary education/primary education or less), smoking status (never/former/current), type 2 diabetes prevalence, osteoporotic fractures prevalence, energy (kcal/day), calcium (mg/day), vitamin D (µg/day), glycemic index, vegetables and fruits (g/day).

**Table 5 nutrients-15-01791-t005:** Association of bone mineral density (T-score) and estimated phytate intake (by tertiles and per each 25 mg/100 kcal).

	Tertile 1 <15.0 mg/100 kcal	Tertile 2 [15.0–28.4] mg/100 kcal	Tertile 3 >28.4 mg/100 kcal	*p*-Value for Trend	Phytate (per 25 mg/100 kcal)	*p*-Value
Femoral Neck, n	185	187	182		554	
Crude Model	0 (ref.)	−0.004 (−0.201–0.193)	0.297 (0.099–0.496)	0.004	0.221 (0.080–0.363)	0.002
Adjusted Model *	0 (ref.)	0.040 (−0.156–0.236)	0.274 (0.072–0.475)	0.008	0.200 (0.058–0.342)	0.006
Femoral Ward’s Triangle, n	185	187	181		553	
Crude Model	0 (ref.)	0.022 (−0.194–0.238)	0.327 (0.109–0.545)	0.004	0.225 (0.070–0.380)	0.004
Adjusted Model *	0 (ref.)	0.053 (−0.165–0.271)	0.319 (0.095–0.544)	0.005	0.218 (0.060–0.376)	0.007
Femoral Trochanter, n	185	187	181		553	
Crude Model	0 (ref.)	−0.109 (−0.313–0.095)	0.138 (−0.067–0.344)	0.192	0.144 (−0.002–0.290)	0.053
Adjusted Model *	0 (ref.)	−0.067 (−0.270–0.137)	0.132 (−0.078–0.342)	0.213	0.134 (−0.014–0.282)	0.075
Total Femur, n	177	184	177		538	
Crude Model	0 (ref.)	−0.094 (−0.293–0.104)	0.156 (−0.044–0.356)	0.128	0.136 (−0.006–0.278)	0.061
Adjusted Model *	0 (ref.)	−0.034 (−0.229–0.161)	0.152 (−0.049–0.353)	0.133	0.130 (−0.011–0.271)	0.071
Lumbar Spine L1–L2, n	148	157	164		469	
Crude Model	0 (ref.)	−0.100 (−0.396–0.196)	0.436 (0.143–0.728)	0.003	0.353 (0.149–0.557)	0.001
Adjusted Model *	0 (ref.)	−0.032 (−0.328–0.264)	0.360 (0.059–0.661)	0.016	0.291 (0.084–0.499)	0.006
Lumbar Spine L1–L3, n	147	157	164		468	
Crude Model	0 (ref.)	−0.057 (−0.359–0.245)	0.421 (0.122–0.720)	0.005	0.331 (0.123–0.539)	0.002
Adjusted Model *	0 (ref.)	0.009 (−0.294–0.312)	0.355 (0.047–0.663)	0.021	0.276 (0.064–0.488)	0.011
Lumbar Spine L1–L4, n	147	156	165		468	
Crude Model	0 (ref.)	0.055 (−0.256–0.365)	0.435 (0.129–0.742)	0.005	0.333 (0.120–0.546)	0.002
Adjusted Model *	0 (ref.)	0.115 (−0.198–0.427)	0.388 (0.071–0.705)	0.015	0.287 (0.069–0.505)	0.010
Lumbar Spine L2–L3, n	167	171	176		514	
Crude Model	0 (ref.)	−0.020(−0.328–0.289)	0.370 (0.064–0.676)	0.017	0.276 (0.060–0.492)	0.012
Adjusted Model *	0 (ref.)	0.055 (−0.256–0.366)	0.343 (0.027–0.659)	0.032	0.248 (0.028–0.469)	0.027
Lumbar Spine L2–L4, n	167	170	177		514	
Crude Model	0 (ref.)	0.0073(−0.241–0.387)	0.389 (0.078–0.699)	0.014	0.296 (0.077–0.515)	0.008
Adjusted Model *	0 (ref.)	0.145 (−0.171–0.462)	0.384 (0.063–0.705)	0.018	0.280 (0.056–0.504)	0.014
Lumbar Spine L3–L4, n	167	170	176		513	
Crude Model	0 (ref.)	0.114 (−0.217–0.444)	0.391 (0.063–0.719)	0.019	0.295 (0.065–0.526)	0.012
Adjusted Model *	0 (ref.)	0.185 (−0.150–0.520)	0.390 (0.049–0.073)	0.025	0.281 (0.044–0.519)	0.020

Linear regression models were used to evaluate the association between T-scores and estimated phytate intake (tertiles and per each 25 mg/100 kcal). Results are expressed as β coefficients (95% CIs). * Models adjusted for age (years), BMI (kg/m^2^), physical activity (MET•min/week), educational level (higher education/technician or secondary education/primary education or less), smoking status (never/former/current), type 2 diabetes prevalence, osteoporotic fractures prevalence, energy (kcal/day), calcium (mg/day), vitamin D (µg/day), glycemic index, vegetables and fruits (g/day).

**Table 6 nutrients-15-01791-t006:** Association of bone mineral density (T-score) in lumbar spine L1–L4 and estimated phytate intake (by tertiles and per each 25 mg/100 kcal) by subgroups.

	Tertile 1 <15.0 mg/100 kcal	Tertile 2 [15.0–28.4] mg/100 kcal	Tertile 3 >28.4 mg/100 kcal	*p*-Value for Trend	Phytate (per 25 mg/100 kcal)	*p*-Value
Age						
≤66 years	72	69	100		241	
	0 (ref.)	0.149 (−0.316–0.615)	0.860 (0.424–1.296)	<0.001	0.580 (0.260–0.901)	<0.001
>66 years	75	87	65		227	
	0 (ref.)	−0.015 (−0.452–0.421)	−0.274 (−0.757–0.210)	0.279	−0.021 (−0.443–0.400)	0.920
*p for interaction*				*0.021*		*0.008*
BMI						
≤32.6 kg/cm^2^	66	82	78		226	
	0 (ref.)	0.081 (−0.370–0.533)	0.185 (−0.283–0.654)	0.432	0.221 (−0.151–0.593)	0.243
>32.6 kg/cm^2^	84	74	87		242	
	0 (ref.)	0.181 (−0.262–0.624)	0.607 (0.168–1.046)	0.007	0.425 (0.080–0.770)	0.016
*p for interaction*				*<0.001*		*<0.001*
Type 2 Diabetes						
No	112	120	121		353	
	0 (ref.)	0.332 (−0.017–0.681)	0.528 (0.170–0.887)	0.004	0.332 (0.083–0.581)	0.009
Yes	35	36	44		115	
	0 (ref.)	−0.511 (−1.209–0.188)	0.078 (−0.583–0.739)	0.697	0.321 (−0.135–0.776)	0.165
*p for interaction*				*0.023*		*0.010*

Linear regression models were used to evaluate the association between T-score L1–L4 and estimated phytate intake (tertiles and per each 25 mg/100 kcal). Results are expressed as β coefficients (95% CIs). Models were adjusted for age (years), BMI (kg/m^2^), physical activity (MET•min/week), educational level (higher education/technician or secondary education/primary education or less), smoking status (never/former/current), type 2 diabetes prevalence, osteoporotic fractures prevalence, energy (kcal/day), calcium (mg/day), vitamin D (µg/day), glycemic index, vegetables and fruits (g/day). *p*-value of interaction with estimated phytate intake was indicated in italics.

## Data Availability

The data that support the findings of this study are available from PREDIMED-Plus trial Steering Committee. Restrictions apply to the availability of these data, which were used under license for this study. Data are available from the authors with the permission of PREDIMED-Plus trial Steering Committee.

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
