# Peer review of "Estimated Phytate Intake Is Associated with Bone Mineral Density in Mediterranean Postmenopausal Women"

_nutrients, 2023, doi:10.3390/nu15071791_

Round 1

Reviewer 1 Report

Overall comments:

The manuscript is well written and interesting. I have two main issues, that should be easily resolved.

Firstly, the proposed mechanism by which phytate has the stated benefit should be more clearly stated in more places in the manuscript. For example, I would expect to see it at least mentioned in the abstract.

Secondly, there is a lack of discussion of phytate as an inhibitor of mineral bioavailability, including calcium. This is currently insufficiently covered by two lines late in the discussion. I suggest this needs to be mentioned in the introduction and covered in more detail in the discussion. I also do not feel that the references used in discussing this points are sufficient, and suggest the below as further material to use.

- ISBN: 9780429134883

- Gibson et al. A review of phytate, iron, zinc, and calcium concentrations in plant-based complementary foods used in low-income countries and implications for bioavailability

- DOI: 10.1080/10408398.2020.1846014

- https://doi.org/10.1002/fsn3.1127

- https://doi.org/10.1016/j.jfca.2019.01.023

Specific comments:

L77-78. More detailed explanation of how this works is needed, not just references.

L314. Explain how. What is the mechanism?

L318. Should this state "reduce bone resorption", rather than "stop"?

L344. Should cereals also be noted here, as well as nuts and legumes?

References: particularly in the discussion, the references cited do not always match with the context of the text (e.g. L346-352). Further, only 60 references are listed in the references section, but numbers above 60 are cited in the text. Check this error.

L348-352. This is not a sufficient explanation of phytates role as an antinutrient, which is very relevant to bone health due to calcium bioavailability. Must be extended and the trade-off between this role and the protective effect you have found discussed.

L365-369. Remove these last three sentences; they are unconventional and add nothing to the manuscript.

Conclusions: I would add your proposed mechanism here too.

Grammatical comments:

L74. comma needed between "acid" and "an"

L77. sentence not right. Suggest removing "basic food of a balanced diet"

L82. Missing "an" before "osteoclastogenesis"

L112. Were, not was

L141. "a dietitian" or "dieticians"

L300. Remove "the"

L303. "has", not "have"

L308. "inhibited"

L324. Remove "body's"

L327-331. Very long sentence with grammatical errors. Suggest rewriting as two sentences.

Throughout the document there are many unnecessary hyphens in the middle of words, likely the result of previous formatting. It needs to be checked throughout for these.

Author Response

We thank to the reviewer for his/her appreciation and constructive comments, which have led to a much-improved version of our manuscript. All his/her comments/suggestions are addressed as detailed in this point-by-point reply and the changes in the manuscript have been marked with control track and highlighted.

Overall comments:

The manuscript is well written and interesting. I have two main issues, that should be easily resolved.

Comment 1#. Firstly, the proposed mechanism by which phytate has the stated benefit should be more clearly stated in more places in the manuscript. For example, I would expect to see it at least mentioned in the abstract.

Response to comment 1#. We thank the reviewer for this comment which help to improve the manuscript. We have added a sentence in the abstract and a graphical abstract explaining the proposed mechanism (see abstract, lines 48-49).

Comment 2#. Secondly, there is a lack of discussion of phytate as an inhibitor of mineral bioavailability, including calcium. This is currently insufficiently covered by two lines late in the discussion. I suggest this needs to be mentioned in the introduction and covered in more detail in the discussion. I also do not feel that the references used in discussing these points are sufficient, and suggest the below as further material to use.

- ISBN: 9780429134883

- Gibson et al. A review of phytate, iron, zinc, and calcium concentrations in plant-based complementary foods used in low-income countries and implications for bioavailability.

- https://doi.org/10.1080/10408398.2020.1846014

- https://doi.org/10.1002/fsn3.1127

- https://doi.org/10.1016/j.jfca.2019.01.023

Response to comment 2#. We completely agree with this comment. As suggested by the reviewer, the role of phytate as an inhibitor of mineral bioavailability is now covered in more detail in discussion section (see lines 355-377). We have also used the suggested references.

Specific comments:

Comment 3#. L77-78. More detailed explanation of how this works is needed, not just references.

Response to comment 3#. We thank the reviewer for this comment which help to improve the manuscript. We have added more details about the mechanism of action of phytate on bone resorption (see introduction section, lines 87-94).

Comment 4#. L314. Explain how. What is the mechanism?

Response to comment 3#. We thank the reviewer for this comment. We have added a paragraph about phytate and their hydrolysates (see discussion section, lines 317-320).

Comment 5#. L318. Should this state "reduce bone resorption", rather than "stop"?

Response to comment 5#. We agree with this comment. We have indicated “reduce bone resorption” rather than “stop” (see line 325).

Comment 6#. L344. Should cereals also be noted here, as well as nuts and legumes?

Response to comment 6#. We thank the reviewer for this comment which help to improve the manuscript. We have indicated “nuts, legumes and cereals” (see line 351).

Comment 7#. References: particularly in the discussion, the references cited do not always match with the context of the text (e.g. L346-352). Further, only 60 references are listed in the references section, but numbers above 60 are cited in the text. Check this error.

Response to comment 7#. We thank the reviewer for this comment which help to improve the manuscript. We have revised and modified the references (see discussion section, lines 351-377).

Comment 8#. L348-352. This is not a sufficient explanation of phytates role as an antinutrient, which is very relevant to bone health due to calcium bioavailability. Must be extended and the trade-off between this role and the protective effect you have found discussed.

Response to comment 8#. We thank the reviewer for this comment which help to improve the manuscript. We completely agree with this comment. As suggested by the reviewer a sentence about phytate as an inhibitor of mineral bioavailability is now covered in more detail in discussion section (see lines 355-377). We have also used the suggested references.

Comment 9#. L365-369. Remove these last three sentences; they are unconventional and add nothing to the manuscript.

Response to comment 9#. We completely agree with this comment. We have removed these three sentences (see line 390).

Comment 10#. Conclusions: I would add your proposed mechanism here too.

 Response to comment 10#. We thank the reviewer for this comment which help to improve the manuscript. We have added a paragraph in conclusion section explaining the proposed mechanism of phytate on bone resorption (see conclusion section, lines 397-399)

Comment 11#. Grammatical comments:

L74. comma needed between "acid" and "an"

L77. sentence not right. Suggest removing "basic food of a balanced diet"

L82. Missing "an" before "osteoclastogenesis"

L112. Were, not was

L141. "a dietitian" or "dieticians"

L300. Remove "the"

L303. "has", not "have"

L308. "inhibited"

L324. Remove "body's"

L327-331. Very long sentence with grammatical errors. Suggest rewriting as two sentences.

Response to comment 11#. We completely agree with these grammatical comments. We have corrected and marked all these errors. We thanks to the reviewer for these appreciations which improve the quality of our paper. The changes have been marked.

Comment 12#. Throughout the document there are many unnecessary hyphens in the middle of words, likely the result of previous formatting. It needs to be checked throughout for these.

Response to comment 12#. We completely agree with this comment. The have revised and removed hyphens throughout the text. The changes have been marked.

Reviewer 2 Report

General:

The authors present substantial evidence that phytate and BMD are correlated, and advocate for controlled trials to test this effect. However, they tread lightly on the fact that phytate decreases the absorption of calcium and other nutrients. More discussion should be included on the potential off-setting effects of increased phytate intake and lower absorption of some nutrients, particularly if diets are low for example, in calcium.

Specific:

Li 110: FFQ – More details are needed as the manuscript relies on understanding phytate intake – how long a period did the FFQ cover, and was it recorded more than once?

Li 349: Absorption: Rather than ‘may’ reduce bioavailability, phytate is known to reduce calcium absorption. This aspect needs to be discussed as it is a counter argument to increased phytate intake (li 375), particularly when calcium intake is low as this could cause other issues (e.g., renal issues, see Kim et al. eLife 2020;9:e52709. DOI: https://doi.org/10.7554/eLife.52709). Specifically, the authors need to consider the effects of decreased absorption of calcium (and other nutrients, such as zinc) when proposing increasing intakes of phytate.

Table 2: Were the years since menopause recorded, and considered in the calculations?

Table 2: The list of nutrients do not include some that are important for bone e.g., P and zinc. Can these be added, or the results provided in the results?

Minor:

Li 45: ‘associated with BMD in women younger… ‘ should that be ‘associated with lumbar spine BMD…’

Li 49: The conclusion should specify which population should be considered for this type of trial – presumably postmenopausal women with high BMI.

Li 87: What is ‘adequate’?

Author Response

We thank to the reviewer for his/her appreciation and constructive comments, which have led to a much-improved version of our manuscript. All his/her comments/suggestions are addressed as detailed in this point-by-point reply and the changes in the manuscript have been highlighted.

General:

Comment 1#. The authors present substantial evidence that phytate and BMD are correlated, and advocate for controlled trials to test this effect. However, they tread lightly on the fact that phytate decreases the absorption of calcium and other nutrients. More discussion should be included on the potential off-setting effects of increased phytate intake and lower absorption of some nutrients, particularly if diets are low for example, in calcium.

Response to comment 1#. We thank the reviewer for this comment which help to improve the manuscript. As suggested by the reviewer, the role of phytate as an inhibitor of mineral bioavailability is now covered in more detail in discussion section (see lines 355-377).

Specific:

Comment 2#. Li 110: FFQ – More details are needed as the manuscript relies on understanding phytate intake – how long a period did the FFQ cover, and was it recorded more than once?

Response to comment 2#. We completely agree with this comment and then, we will clarify this point. The FFQ collected information about the food intake during the previous year. The FFQ is repeated yearly in PREDIMED-Plus participants, but given that this is a cross-sectional study using baseline data, only data of the first FFQ - administered at recruitment - is presented. We have added this paragraph in the manuscript (See subsection “Estimated Phytate Intake…”, lines 137-140).

Comment 3#. Li 349: Absorption: Rather than ‘may’ reduce bioavailability, phytate is known to reduce calcium absorption. This aspect needs to be discussed as it is a counter argument to increased phytate intake (li 375), particularly when calcium intake is low as this could cause other issues (e.g., renal issues, see Kim et al. eLife 2020;9:e52709. DOI: https://doi.org/10.7554/eLife.52709). Specifically, the authors need to consider the effects of decreased absorption of calcium (and other nutrients, such as zinc) when proposing increasing intakes of phytate.

Response to comment 3#. As suggested by the reviewer, the role of phytate as an inhibitor of mineral bioavailability is now covered in more detail in discussion section (see lines 355-377). We have also used the recommended bibliography.

Comment 4#. Table 2: Were the years since menopause recorded, and considered in the calculations?

Response to comment 4#. Thanks to the review for this comment. We have recorded “menopausal age (years)” and “time since menopause (months)” and considered for the calculations. Neither of the two variables was correlated with phytate intake or with BMD. We have included the variable “Menopausal age (years)” in table 2.

Comment 5#. Table 2: The list of nutrients do not include some that are important for bone e.g., P and zinc. Can these be added, or the results provided in the results?

 Response to comment 5#. Thanks to the review for this comment. We have added P and Zinc in table 2. Women in the top phytate tertile consumed more P and Zn than those in the lowest phytate tertile (see table 2).

Minor:

Comment 6#. Li 45: ‘associated with BMD in women younger… ‘ should that be ‘associated with lumbar spine BMD…’

Response to comment 6#. Thanks to the review for this comment. We have indicated “associated with lumbar spine BMD” (see line 47).

Comment 7#. Li 49: The conclusion should specify which population should be considered for this type of trial – presumably postmenopausal women with high BMI.

Response to comment 7#. We thank the reviewer for this comment which help to improve the manuscript. We have now indicated the population for the trial (see line 51).

Comment 8#. Li 87: What is ‘adequate’?

Response to comment 8#. An adequate consumption of phytate is considered around 0.5-1g/day of phytate. It is important to consider that this amount of phytate is ingested in most of the healthy and balanced diet such as Mediterranean diet [26]. Interestingly, a recent paper has indicated that a phytate consumption higher than 307 mg/day was associated with a normal lumbar BMD (t-score > −1) in post-menopausal women [22]. We have added this information in the discussion section (see lines 373-377) and indicated in the introduction section (see line 100).
